# Establishment and range expansion of *Dermacentor variabilis* in the northern Maritimes of Canada: Community participatory science documents establishment of an invasive tick species

**Andrea M. Kirby, Ellis P. Evans, Samantha J. Bishop, Vett K. Lloyd** [ORCID] *

Dept. Biology, Mount Allison University, Sackville, New Brunswick, Canada

* vlloyd@mta.ca

**Data Availability Statement:** All tick mapping data is available at https://gnb.socrata.com/Health-and-

## Abstract

Tick populations are dependent on a complex interplay of abiotic and biotic influences, many of which are influenced by anthropic factors including climate change. *Dermacentor variabilis*, the wood tick or American dog tick, is a hardy tick species that feeds from a wide range of mammals and birds that can transmit pathogens of medical and agricultural importance. Significant range expansion across North America has been occurring over the past decades;this study documents northwards range expansion in the Canadian Maritime provinces. Tick recoveries from passive surveillance between 2012 and 2021 were examined to assess northward population expansion through Atlantic Canada. At the beginning of this period, *D. variabilis* was abundant in the most southerly province, Nova Scotia, but was not considered established in the province to the north, New Brunswick. During the 10-year span covered by this study, an increasing number of locally acquired ticks were recovered in discrete foci, suggesting small established or establishing populations in southern and coastal New Brunswick. The pattern of population establishment follows the climate-driven establishment pattern of *Ixodes scapularis* to some extent but there is also evidence of successful seeding of disjunct populations in areas identified as sub-optimal for tick populations. Dogs were the most common host from which these ticks were recovered, which raises the possibility of human activity, via movement of companion animals, having a significant role in establishing new populations of this species. *Dermacentor variabilis* is a vector of several pathogens of medical and agricultural importance but is not considered to be a competent vector for *Borrelia burgdorferi*, the etiological agent of Lyme disease; our molecular analysis of a subset of *D. variabilis* for both *B. burgdorferi* and *B. miyamotoi* did not confirm any with *Borrelia*. This study spans the initial establishment of this tick species and documents the pattern of introduction, providing a relatively unique opportunity to examine the first stages of range expansion of a tick species.

Wellness/Tick-Data-2012-to-2018-Donn-es-relatives-aux-tique/3mpw-72pb

**Funding:** Seed funding for the tick bank was provided by the Canadian Lyme Disease Foundation to VKL (CanLyme 2014-1, https://canlyme.com/). Operational funding was provided by the Natural Sciences and Engineering Research Council to VKL (NSERC 4426-2015, https://www.nserc-crsng.gc.ca) as well as private donations for tick research. EPE was supported by a Undergraduate Student Research Award from the Natural Sciences and Engineering Research Council (NSERC USRA). The funders had no role in study design, data collection and analysis, decision to publish, or preparation of the manuscript.

**Competing interests:** The authors have declared that no competing interests exist.

## Introduction

The stability and range of animal populations are dependent on the complex interplay between biotic and abiotic factors; ticks are no exception [1–5]. Hard ticks of the family *Ixodidae* consist of 14 genera, including *Dermacentor*, of which there are approximately 40 species including several found in North America [6]. *Dermacentor variabilis*, colloquially referred to as the wood tick or the American dog tick, is abundant throughout much of continental USA, Mexico and southern Canada [1, 7, 8]. The range of this species has been expanding both northward into Canada and west and east across the continent [1, 7, 8]. Different tick species have an optimal combination of climactic and biological factors that influence their ability to complete their life cycle and reproduce in any given environment [1, 5, 9]. Climactic factors influencing the range of this species include rainfall, duration and magnitude of seasonal extreme temperatures, and related climactic variables. Biotic factors include land cover, host availability and host dispersal [1, 10, 11]. While ticks do not move great distances independently, they feed for several days in each life stage so infestation of a bird or large mammal can result in long-range dispersal. *Dermacentor variabilis* feeds from a wide range of hosts [7] so the movements of these hosts can promote wide dispersal of the tick. Human activity can affect all of the factors regulating tick distribution. Additionally, *D. variabilis* are among the most commonly found ticks on humans [6, 12] and are frequently found on dogs and agricultural animals so, movement of humans, companion animals or agricultural animals may significantly contribute to tick range expansion.

The large geographic range of *D. variabilis* and the multiple pathogens that it vectors makes this species one of the most economically and medically important tick species in North America [1]. *Dermacentor variabilis* is a known vector of multiple pathogens. These include *Rickettsia rickettsii*, the etiological agent of Rocky Mountain spotted fever [13], and more commonly, other pathogenic spotted fever-producing *Rickettsia* species [13] and some related *Ehrlichia* species such as *Ehrlichia canis* [14]. Additionally, *D. variabilis* can vector *Francisella tularensis*, the cause of tularemia [15, 16], other parasites including *Babesisa* sp. [17], known or suspected pathogenic viruses [18–20], and additionally can cause tick paralysis disease [21, 22]. *D. variabilis* has been found to be infected with *Cytauxzoon felis*, which causes serious feline disease [23, 24], *Anaplasma phagocytophilum* and *A. marginale* [25, 26], which cause human and bovine anaplasmosis, respectively, and *E. chaffeensis* and *E. ewingii*, agents of human ehrlichiosis in some [26–28], but not all [29], cases. Similarly, *Borrelia burgdorferi*, one of the causative agents of Lyme disease, has been identified in this and related *Dermacentor* species [26, 30–32]. However, for a tick to transmit rather than simply acquire a pathogen is more difficult to determine experimentally. At least for *B. burgdorferi*, previous research suggests that *Dermacentor* spp. ticks are not competent vectors of *B. burgdorferi* [33–37], and assumed by extension to not be competent vectors for related *Borrelia* species such as *Borrelia miyamotoi*. Nevertheless, incidences of tularemia [38], bovine anaplasmosis [39], and Rocky Mountain spotted fever [40] have worsened over recent decades in the United States. *Francisella tularensis* is endemic in many parts of Canada, although cases of tularemia in Canada remain rare and are thought to arise from direct exposure to the pathogen rather than the tick vector [41]. National information on the occurrences of spotted fevers and anaplasmosis is not readily available, although evidence from regional studies suggests that they may be rare but increasing [7, 8]. As *D. variabilis* populations expand in scope and numbers, increased encounters between humans, companion and agricultural animals will necessarily increase, making surveillance of this species of importance for both economic and human and animal health.

Nova Scotia and New Brunswick are eastern maritime Canadian provinces, with Nova Scotia lying to the south of New Brunswick. New Brunswick additionally shares a land border

with the province of Quebec and the state of Maine of the USA. Relative to its neighbors, the climate of New Brunswick is cooler, with more protracted periods of snow cover and there is widespread low-elevation forested land with extensive waterways and wetlands, habitats that foster *D. variabilis* populations in other regions [42–46]. *Dermacentor variabilis* has been documented in Nova Scotia since the 1940s [42, 47]. It is reported that the tick was introduced in the early 1900s on sporting dogs from the United States [47], although population establishment by other animals or humans cannot be excluded. Since that time, the species has continued to expand its range throughout the province [42, 48]. At the start of this study, *D. variabilis* was abundant throughout Nova Scotia, but had not been recovered by active tick surveillance or from non-traveled hosts in New Brunswick [12, 49]. Suitable, if not optimal, climates for this tick species are present throughout much of the Canadian Maritimes, excluding northern New Brunswick, and are predicted to increase in suitability with further climate change [1, 50, 51]. Thus, it is reasonable to anticipate that *D. variabilis* will establish populations in New Brunswick that will subsequently expand. If distribution and expansion is driven by dispersal by wildlife hosts, the spread would likely be continuous and responsive to climate and other abiotic variables, as appears to be the case for *Ixodes scapularis*, a tick species that invaded the province earlier [50, 51]. In contrast, if introductions of the ticks is driven primarily by anthropic factors such as the movement of humans and companion animals the pattern of tick recoveries would be more dispersed and discontinuous, reflecting the location of introductions.

To monitor and better understand the spread of this tick species, we have analysed 10 years of tick recovery data from New Brunswick. These data were obtained from a large community science initiative in which members of the public participated in passive tick surveillance by submitting ticks found on themselves, on companion animals and in the environment, in exchange for tick identification and pathogen testing. We document the increasing local acquisition of *D. variabilis* from multiple discontinuous foci across the province. These foci were initially seen in the south and coastal regions with milder ambient temperatures, and later in the less temperate northern regions, suggesting population expansion is controlled both by climate and multiple independent introductions by human activity. Thus, this study spans the period of initial establishment in a region, providing a relatively unique opportunity to examine the first stages of range expansion of a tick species.

## Materials and methods

### Specimen acquisition

The Mount Allison Tick Bank serves as an archive of ticks and their DNA, collected through passive surveillance, primarily from New Brunswick (NB) and the other Canadian maritime provinces, Nova Scotia (NS) and Prince Edward Island (PEI). Ticks are donated by the general public, primarily via veterinary clinics but also directly from the public. When the tick bank was initiated, ticks were solicited from veterinary clinics by letter and email and presentation at the annual regional veterinary medicine conference (2012 and 2013). No organized promotion of the tick bank occurred in subsequent years although it was mentioned in sporadic media reports; engagement by veterinary clinics was fairly consistent throughout the span of the study.

Upon receipt, *Ixodes* spp. ticks were identified to species, life stage, sex, and state of engorgement based on [52, 53] (University of Rhode Island, www.tickencounter.org). The initial focus of the tick bank was on *Ixodes* species ticks, so all non-*Ixodes* ticks were photographed and archived; identification to species, when possible, occurred at a later date based on images and examination of archived residual specimens when possible. Each sample is

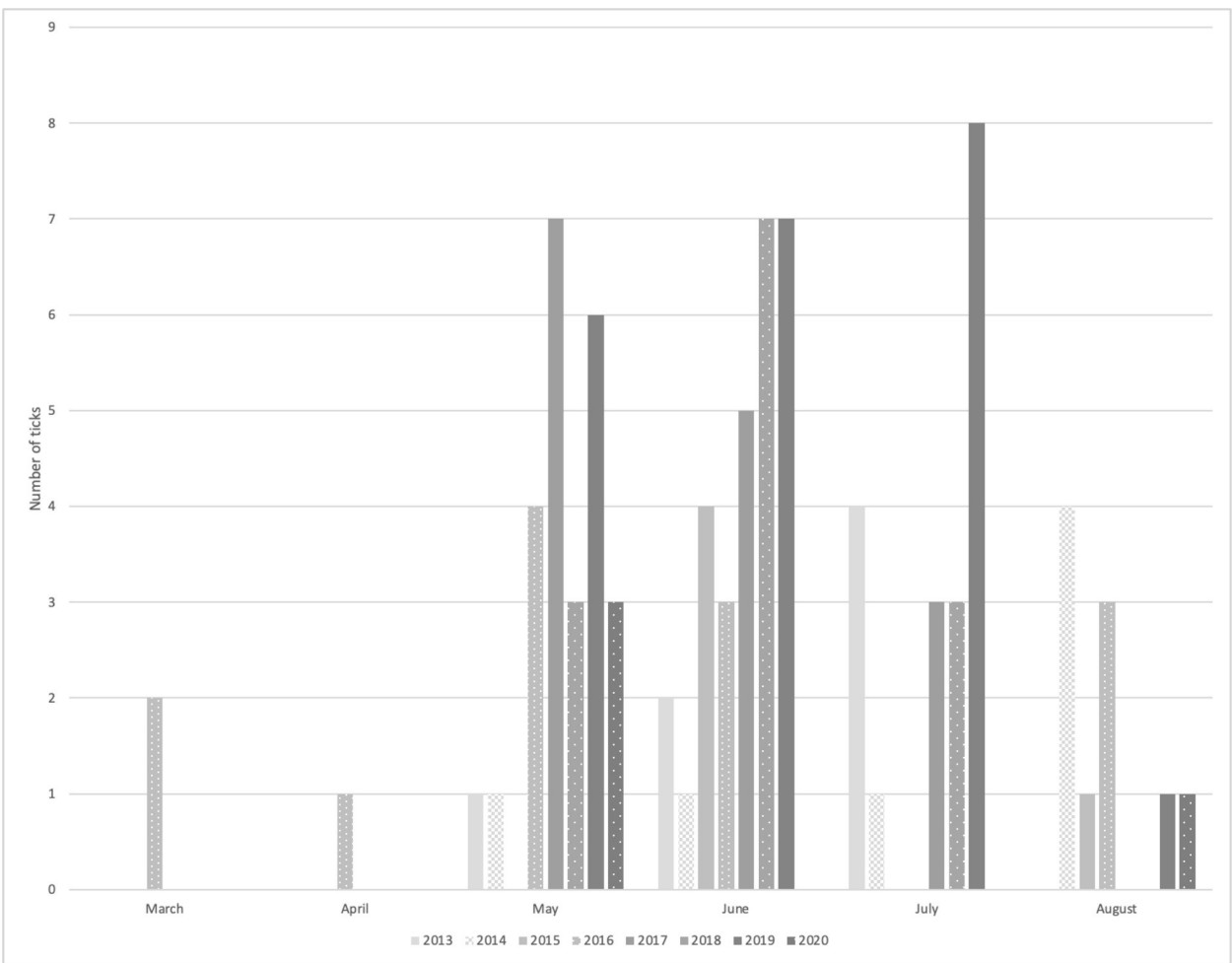

**Fig 1. Locally-acquired *Dermacentor variabilis* recoveries by passive surveillance, by month, in New Brunswick from 2013 to 2020.**

accompanied by a submission form which included the date and location where the tick was encountered, instances of recent travel, whether the tick was attached, the species of the host, and contact and consent information. Photographs of the mouthparts, dorsal aspect, and ventral aspect for each tick were taken under a Leica EZ4D dissecting microscope. Ticks were subsequently longitudinally bisected (with the exception of larvae and unengorged nymphs) with half used for DNA extraction and subsequent testing and the other half archived at -20˚C. This study selected those ticks which were identified as *Dermacentor* spp. collected in NB between 2012 and 2021. Any disclosed out-of-province travel in the two weeks prior to tick discovery was noted. These ticks were excluded from mapping (Fig 2), but not molecular analysis. Funding considerations prompted the conversion of free tick testing to a user-pay service in 2020, drastically reducing tick submissions, particularly for non-*Ixodes* species. An additional four tick records from 2021 were provided by a tick testing company, Geneticks [54], and 58 records of Dermacentor species with no documented travel history submitted for identification to the tick surveillance platform, eTick [48].

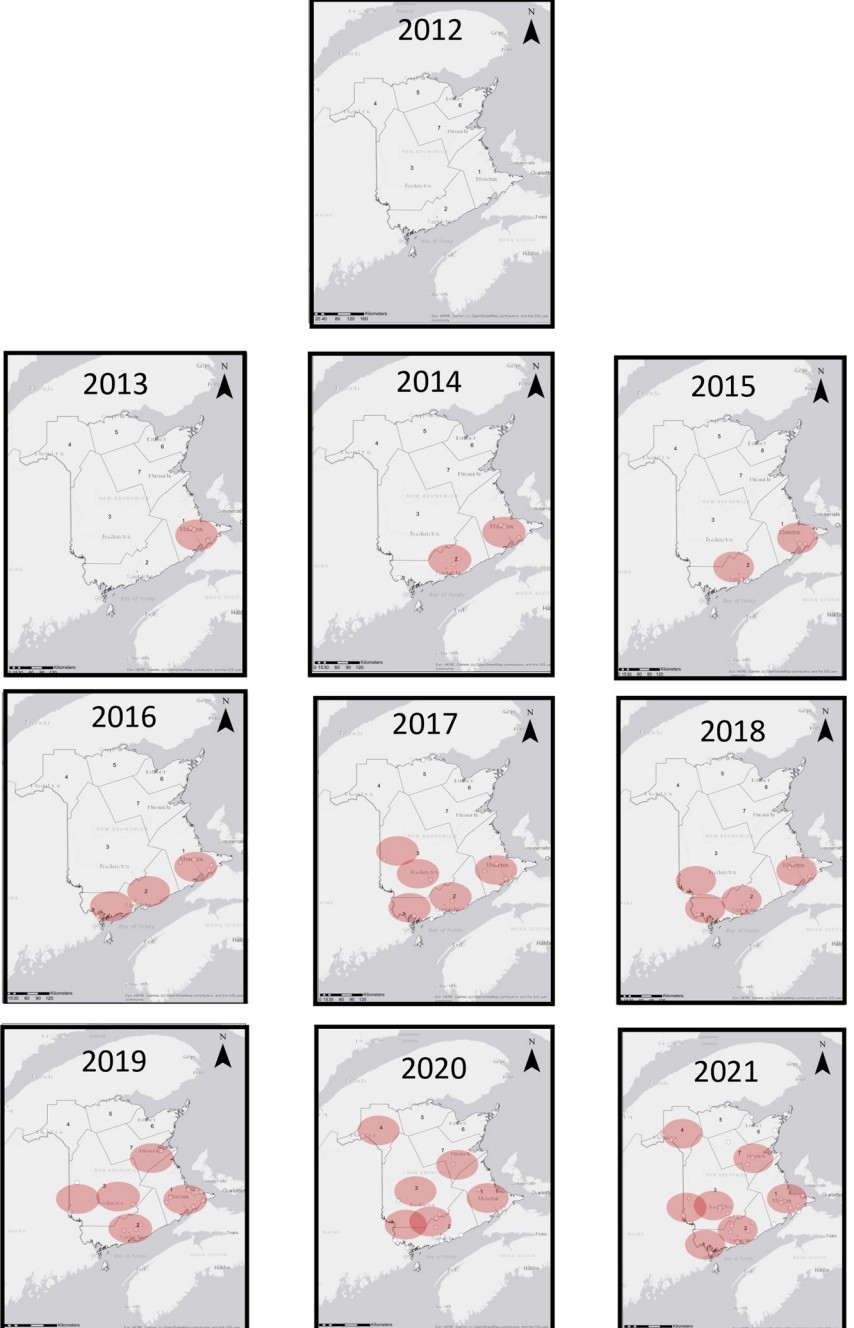

**Fig 2. The distribution of locally-acquired *D. variabilis* ticks from New Brunswick from 2012–2021.** Areas of concentration, 50km in diameter, of collection lasting > 2 years, are indicated with red circles. The GIS map template includes human health zones and scale. Map templates were obtained from ArcGIS Online maps hosted by Esri.

## Mapping and statistical analysis

Mapping of the location of the tick recoveries used GIS map templates that are the intellectual property of Esri and are used herein under license (Copyright © 2020 Esri and its licensors. All rights reserved).

**Table 1. Primers and conditions used to amplify *Borrelia* DNA.**

| Primer | Target gene | Sequence (5' → 3') | Amplicon length (bp) | Melting temperature (˚C) | Annealing temperature (˚C) |
|---|---|---|---|---|---|
| FlaB out L | *Borrelia burgdorferi FlaB* | GCATCACTTTCAGGGTCTCA | 350 | 62.9 | 55 |
| FlaB out R | | TGGGGAACTTGATTAGCCTG | | 63.9 | |
| FlaB in L | | CTTTAAGAGTTCATGTTGGAG | | 55.6 | 58 |
| FlaB in R | | TCATTGCCATTGCAGATTGT | | 64.1 | |
| OspA out L | *Borrelia burgdorferi OspA* | CTTGAAGTTTTCAAAGAAGAT | 521 | 54.4 | 55 |
| OspA out R | | CAACTGCTGACCCCTCTAAT | | 60.7 | |
| OspA in L | | ACAAGAGCAGACGGAACCAG | | 64.4 | 58 |
| OspA in R | | TTGGTGCCATTTGAGTCGTA | | 64.1 | |
| 23S out F | *Borrelia* spp. *23S rRNA* | GTATGTTTAGTGAGGGGGGTG | 587 | 57.72 | 50 |
| 23S out R | | GGATCATAGCTAGGTGGTTAG | | 57.74 | |
| 23S in F | *Borrelia burgdorferi 23S rRNA* | ATGTATTCCATTGTTTTAATTACG | 340 | 52.22 | 51 |
| 23S in R | | GACAAGTATTGTAGCGAGC | | 53.66 | |
| 23S Bmiya Fin | *Borrelia miyamotoi 23S* forward inner | ATAAACCTGAGGTCGGAGG | 447 | | 60 |
| 23S Bmiya Rin | *Borrelia miyamotoi 23S* reverse inner | AAAGTGTGGCTGGATCACC | | | |

To assess the probability of the increased *D. variabilis* recoveries occurring by chance, both over time and in northern versus southern regions, chi squared analysis was used. The four years 2013–2016 (inclusive) were used as the "early" years and the four years 2017–2020 were used as "late" years. Recoveries over multiple years were combined to reduce random fluctuation in the numbers of rare recoveries. The year 2021 was excluded as submissions of all non-*I. scapularis* ticks decreased with the imposition of cost-recovery for tick testing; data from the year 2012 was excluded for symmetry. For geographic analysis, the province was divided into southern (health zone 1 and 2), middle (health zones 3 and 7) and northern regions (health zones 4 to 6). These regions were based on human health zones [55] rather than ecological variables as anthropic information important for passive surveillance are categorized using these health zones, but they do correspond reasonably well with geographic and climactic parameters [50].

**Table 2. Summary of *Dermacentor variabilis* recoveries from New Brunswick by year.**

| | Total | Tick life stage | | | host | | |
|---|---|---|---|---|---|---|---|
| year | for year[1] | adult female | adult male | larvae/nymph | dog | human | other[2] |
| 2012 | 0 | 0 | 0 | 0 | 0 | 0 | 0 |
| 2013 | 7 | 6 | 1 | 0 | 6 | 1 | 0 |
| 2014 | 7 | 6 | 1 | 0 | 7 | 0 | 0 |
| 2015 | 5 | 4 | 1 | 0 | 5 | 0 | 0 |
| 2016 | 11 | 10 | 1 | 0 | 4 | 5 | 2 |
| 2017 | 15 | 5 | 10 | 0 | 10 | 4 | 1 |
| 2018 | 13 | 7 | 6 | 0 | 8 | 5 | 0 |
| 2019 | 24 | 12 | 12 | 0 | 16 | 6 | 2 |
| 2020 | 17 | 9 | 5 | 3 | 5 | 7 | 5 |
| 2021 | 42 | 30 | 12 | 0 | 7 | 30 | 5 |
| total | 141 | 89 | 49 | 3 | 68 | 58 | 15 |

1. Total ticks recovered for that year. Adjacent columns provide breakdown by life stage and host.

2. Other included wildlife (deer, coyote), "walking ticks" - ticks found prior to feeding, either in the outside environment, on clothes or in the house.

## Molecular analysis

All ticks, regardless of species and including those too damaged to identify to species, were tested upon receipt for the presence of 1–3 *B. burgdorferi* genes, *Outer Surface Protein A* (*OspA*), *Flagellin B* (*FlaB*) and the *23S ribosomal RNA* (*rRNA*) gene (Table 1; [56, 57]). Presumably due to competition from tick DNA, if the abundance of the *B. burgdorferi* target DNA was low, in some cases not all *Borrelia* primers sets would produce an amplicon. This was not specific to *D. variabilis* as it was also noted for *I. scapularis* and *I. cookei* ticks tested in this manner [12, 58]. Extraction of DNA, amplification and visualization were completed as described by [59]. For all nested PCR (nPCR) reactions, the reaction volumes and nPCR protocol were as described in [59] with the annealing temperatures given in Table 1. Negative controls consisting of water in place of DNA were conducted at the start and end of each experimental PCR manipulation in the same workspace, in order to detect reagent contamination prior to or during nPCR or aerosolization of DNA in the PCR workspace. Tick DNA extraction, PCR manipulations and gel electrophoresis were all conducted in separate locations with independent air flow. Amplicons of sizes indicative of either *Borrelia* spp. or *B. burgdorferi* were submitted for Sanger sequencing at Génome Québec, McGill University (Montréal, QC). The resulting chromatograms were trimmed and manually curated for ambiguous nucleotides using FinchTV and a Nucleotide Blast of each sequence was performed using the National Center for Biotechnology Information's (NCBI) GenBank.

## Results

### *Dermacentor* recoveries

Between 2012 and 2021, 668 *Dermacentor* spp. ticks were collected through community-based passive surveillance in New Brunswick and submitted to the MTA Tick Lab. For 2020 and 2021, ticks and or records of 4 additional *D. variablis* ticks from the tick testing company, Geneticks [54] and 58 from the tick surveillance platform, eTick [48] were obtained. 141 specimens were retained after removal of submissions where the tick was acquired outside of New Brunswick, submissions that did not explicitly exclude travel outside of New Brunswick, ticks too damaged for species-level identification and *D. albipictus* (of which 13 were found from one moose and three dogs; Table 2).

### *Dermacentor variabilis* collected in New Brunswick

Of the submitted *D. variabilis* from New Brunswick, 98.5% (139/141) were adults. In comparison, from the neighboring province of Nova Scotia with extensive and well-established populations, 99.2% were adults (S1 Table in S1 File). Of the adult ticks, approximately 10–20% were male in the first years (2012–2016); from 2017 onwards, the proportion of males recovered increased to approximately 30–60% (Table 2). Overall, dogs were the most common host (48%), with humans also well represented (41%), although canine-derived samples were, as expected, under-represented after 2020 when free submissions were discontinued. *D. variabilis* were also readily found on wildlife, or walking in the outside or inside environment (Table 2), consistent with their wide host range [7]. The major peak of *D. variabilis* tick recoveries was from May to August (Fig 1).

There is a trend of increasing *D. variabilis* recoveries over time evident by inspection and supported by comparing the proportion of *D. variabilis* to total tick recoveries for years 2017–2020 to 2013–2016 by Chi squared analysis (p = 0.014). To determine if this might be because *D. variabilis* were establishing populations in the province, the geographical origins of locally acquired ticks were plotted (Fig 2). *Dermacentor variabilis* specimens collected in 2012 were all

were associated with travel to regions with well-established and abundant *D. variabilis* populations (primarily Nova Scotia) [12]. Starting in 2013, locally-acquired *D. variabilis* were recovered in New Brunswick, although close to the New Brunswick-Nova Scotia border. This was followed in 2014 with locally-acquired ticks found in the southwest of the province in the St. John region, and by 2016, stretching along the southern portion of the province. In 2017, multiple disperse local recoveries were reported. These included recoveries from the southern and middle portions of the province and along the Maine, US border. Widely distributed and expanding populations of *D. variabilis* had been previously documented in Maine, including near the Maine-New Brunswick border, by 2006 [45]. By 2019, locally-acquired *D. variabilis* were additionally being reported along the coastal region. By 2020 *D. variabilis* were also recovered in the northwest of the province at the New Brunswick-Maine-Quebec border; *D. variabilis* having also been reported from this part of Quebec in 2017 [60]. Of note, this widespread recovery occurred during a time when non-essential travel across provincial borders was severely restricted in response to the COVID-19 pandemic. By 2021, *D. variabilis*, while still not abundant, was being reported throughout most of the province of New Brunswick. Chi squared analysis indicated that for both the "early expansion" period of 2013–2016 and the "later expansion" period of 2017–2020, proportion of the *D. varibilis* relative to total tick recoveries in the middle and northern parts of the province were greater than for the southern part of the province ($p = 3.9 \times e^{-7}$ and $1.8 \times e^{-5}$, respectively), where *I. scapularis* was well established and a strong contributor to total tick recoveries.

### *Borrelia burgdorferi* and *B. miyamotoi* in *D. variabilis* ticks

All ticks, regardless of species and including those too damaged to identify to species, were tested upon receipt for 1–3 *B. burgdorferi* genes. Twenty *D. variabilis* ticks showed an amplicon consistent with the potential presence of *B. burgdorferi*. Sequence conformation of re-amplified amplicons from the four of these ticks with at least two amplicons consistent with *B. burgdorferi* genes, produced only one, a *D. variabilis* male (S1 Fig in S1 File), obtained in 2021 from a dog from southeastern New Brunswick without a recent travel history, with a sequence consistent with the *B. burgdorferi 16-23S intergenic spacer (IGS)* region; sequence analysis of the *B. burgdorferi Outer Surface Protein A* gene segment produced uninterpretable sequence (S2 Fig in S1 File). The other three ticks produced uninterpretable sequences.

Screening of ticks for the relapsing fever spirochete, *B. miyamotoi* was not routinely performed upon receipt of the ticks so a total of 181 *D. variabilis* ticks (144 ticks collected in 2017, 34 ticks collected in 2018 and 23 from 2012–2016) were tested. In contrast to the positive control (DNA from the liver of a *B. miyamotoi*-infected *Peromyscus leucopus*) only two *D. variablis* ticks, both adult females from a canine host collected in 2014, produced amplicons consistent with the size expected for *B. miyamotoi*, however, the presence of *B. miyamotoi* could not be confirmed by Sanger sequencing.

## Discussion

The expansion of the geographic range of tick populations, including those of *D. variabilis*, is being increasingly described in various parts of the world [1–4, 7]. Recovery of locally acquired *D. variabilis* increased in the Canadian Maritime province of New Brunswick over the 10-year span of this study; no locally acquired ticks were found when the study started in 2012 but by 2021, ticks were recorded from seven "hot spots" across the region (Fig 2; [48].

## Does the presence of locally acquired ticks mean that self-sustaining populations *D. variabilis* have been established?

The presence of ticks could be explained by adventitious ticks translocated from adjacent high tick population areas. If these ticks were able to survive but were unable to find hosts from which to feed, find a mate or have surviving progeny, then local populations would not be able to establish. There are several lines of evidence that argue for the ticks recovered in this study representing newly established populations. The most obvious argument is that the number of tick recoveries increased throughout the time span captured in this study; populations of *D. variabilis* were both high and stable in the adjacent region of northern Nova Scotia since at least 2014 [12], so if the ticks recovered in New Brunswick were purely adventitious ticks, their number should have also remained stable. Additional evidence that *D. variabilis* recoveries represent the newly established populations include that the ticks were recovered from wildlife hosts and the recovery of immature stages in at least one year. Furthermore, *D. variabilis* were still recovered from multiple locations throughout the province in 2020 when provincial borders were closed to non-essential travel, and these locations were the same regions from which ticks were recovered in previous and subsequent years. The recovery of increased numbers of ticks and recoveries from dispersed locations are all consistent with the establishment of new populations, as well as with the biology of the species. With a broad host range [7] finding hosts should not be a challenge. Finding a mate is a problem that is alleviated one generation after an adult female is translocated while feeding on a host, as long as her eggs are able to hatch and the progeny thrive. As this species is a hardy one with relatively few climactic barriers [61, 62], propagation should not be a limiting consideration. To determine if the *D. variabilis* recoveries reported here do represent newly established populations, it would be helpful to recover additional immature developmental stages, ideally on wildlife, through active surveillance. As [63] did not recover *D. variabilis* specimens during active surveillance in the region during the initial time span covered by this study, 2014, repeated active surveillance would provide valuable conformation. [7] report a good correlation between active and passive surveillance in detecting northward population spread for *D. variabilis* and [8] document population expansion of *D. variabilis* in central Canada using passive surveillance, so the increasing tick recoveries in New Brunswick, reported here is likely to represent the establishment of new *D. variabilis* populations.

Recoveries of *D. variabilis* occurred from multiple locations within the province, initially in the southeast and southwest. By 2014 recoveries started to occur between these foci and in the north of the province. The pattern of *D. variabilis* recoveries, in the milder south and coastal regions where *I. scapularis* first established [50], is suggestive of establishing, and possibly established, populations in the more climatically moderate regions. However, the locally acquired *D. variabilis* did not all originate from the more climatically moderate regions of the province, unlike what was found for *I. scapularis* [50]. Climate modelling by [1] predicts the northward expansion of suitable conditions for *D. variabilis* populations under all climate change scenarios; these suitable regions encompass both Nova Scotia and New Brunswick. Nevertheless, their climate model indicates that both provinces are only marginally suitable for *D. variabilis* populations. This prediction is seemingly at odds with the abundant recoveries of *D. variabilis* in Nova Scotia [12, 48] coupled to the growing recoveries of *D. variabilis* in New Brunswick documented here. However, the model used captured climactic variables but not biotic factors such as host movement that can lead to tick introductions. Additionally, the sensitivity of such models is greater for predicting where tick populations would not be sustainable than where populations would be sustainable, neither does the model predict tick abundance [1]. *D. variabilis* is a tough and versatile tick, capable of survival even in ecosystems

not thought to be suitable such as the challenging climate of northern Canadian Prairies [7, 61, 62] so climactic conditions may not constrain population establishment as significantly as for other tick species.

## How are *D. variabilis* populations being established?

Inspection of the locations of locally-acquired *D. variabilis* in New Brunswick shows that the ticks are found in distinct and discontinuous foci, in this study, generally separated by 100 - 200 km. This finding is consistent with the presence of disjunct populations in both Canada [42, 46, 47] and in the United States. These disjunct populations have been assumed to represent separate tick introductions via infested livestock or companion animals [42, 46, 47]. Perhaps not surprisingly given the common name of "American dog tick", adult *D. variabilis* ticks are the predominant *Dermacentor* species found on companion animals in the United States [64] and specimens collected in our tick bank were frequently recovered from dogs (Table 2, [12]. Immature *D. variabilis* have been found feeding from small mammals [65–67] but rarely from birds or humans [68–71]. This would limit the ability of this species to disperse large distances as juveniles. Adult *D. variabilis*, however, readily feed from larger mammals including humans, companion and agricultural animals. This raises the obvious possibility that the distributed pattern of *D. variabilis* recoveries represent anthropic introductions mediated by the travel of companion dogs, a mode of introduction that has been previously implicated in the introduction of *D. variabilis* in other regions [42, 46, 47]. If population establishment and expansion of *D. variabilis* is more reflective of chance anthropic "seedings" by human actions than large scale climactic factors, risk modeling will be complex. Finer scale population surveillance that captures anthropic variables, biotic and abiotic factors that include human use of habitats, microclimates, quantification of tick density and surveillance of wildlife for the pathogens transmitted by *D. variabilis*, among other factors, is warranted to monitor and predict tick populations to protect human and animal health in regions of expanding tick populations.

## *Dermacentor variabilis* and *Borrelia*

While *D. variabilis* can transmit multiple pathogens of human and veterinary importance, their ability to maintain and transmit *Borrelia* spirochetes remains contentious with conflicting reports. The possibility of *D. variabilis* ticks acquiring *Borrelia* spirochetes from infected hosts is not surprising, and does not necessarily mean that the ticks could transmit the pathogen. There have been studies that have detected *Borrelia* spirochetes in *D. variabilis* that fed from infected hosts. *D. variabilis* larvae removed from infected *P. leucopus* in a highly endemic area had *B. burgdorferi* spirochetes identified both serologically and morphologically [30]. Similarly, [32] found *B. burgdorferi* in adult *D. reticulatus* in Belarus; *B. burgdorferi*, *B. afzelii*, or *B. valaisiana* were identified in 2.7% of the 226 ticks tested. There also have been isolated reports of *Dermacentor* spp. ticks being infected with *B. burgdorferi* elsewhere in North America. An adult *D. albipictus* found feeding on a canine host in Ontario, Canada had molecularly identified *B. burgdorferi*, although unfortunately the dog was not tested for *B. burgdorferi* exposure to investigate prior infection or transmission [31]. The finding of a potentially *Borrelia*-positive reported *D. variabilis* may fall into this category of a *Dermacentor* tick acquiring *B. burgdorferi* from an infected dog, although again the dog was not tested for *Borrelia*. While there is evidence of *D. variabilis* carrying *B. burgdorferi*, caution needs to be exercised in interpreting these results. In all cases, amplicons suggestive of *B. burgdorferi* require sequence validation to confirm the presence of *B. burgdorferi*. By convention, amplification and sequence conformation of two or more *Borrelia* genes is considered necessary to consider a tick a "true" positive [12, 56], so the *D. variabilis* identified here would not be considered to be a true

positive. This assessment criteria is designed to reduce false positives, although it can also produce false negative results. Competition from the more abundant tick DNA can reduce detection of *Borrelia* sequences and target DNA in archived samples is subject to DNA degradation. Conversely, PCR, the method used in most recent studies to assess the presence of *Borrelia*, can detect DNA from both viable and non-viable, but not yet eliminated, *Borrelia* leading to false positive results. [57] report that *B. burgdorferi* and *B. miymotoi* were detected in approximately 4% and 2%, respectively, of wildlife species from southern New Brunswick. If *D. variabilis* were efficient at acquiring and retaining *Borrelia* infection, approximately 6 and 4, respectively, of the ticks tested in this study would have been expected to be carrying these *Borrelia* species. That fewer were recovered may represent degradation of *Borrelia* DNA in vivo in *D. variabilis* ticks; assessing ticks collected by active surveillance, so possibly sooner after feeding, would address this question.

While *Borrelia* has been detected in *D. variabilis*, this does not necessarily indicate that these ticks can transmit the pathogens and previous research suggests that *Dermacentor* spp. ticks are not competent vectors of *Borrelia burgdorferi*. Their poor vectoral capacity is thought to result from the action of antimicrobial peptides and lysozymes in the tick's haemolymph that are effective in eliminating *B. burgdorferi* [72, 73]. The effectiveness of this immune response was demonstrated in experiments in which *B. burgdorferi* was injected directly into the haemolymph of *D. variabilis*, causing a rapid surge in antimicrobial peptides, lysozymes, and haemocytes leading to rapid bacterial elimination [33, 34]. Consistent with this finding, *D. variabilis* nymphs were found to lack detectable *B. burgdorferi* despite molting from infected larvae and these nymphs were unable to transmit *B. burgdorferi* to naïve mice in the laboratory [37]. These findings are mirrored by evidence that *Dermacentor* ticks appear unable to maintain a *B. burgdorferi* infection in nature [35, 36]. Both *I. scapularis* and *D. variabilis* larvae and nymphs were found to frequently feed on white-footed mice *(Peromyscus leucopus)*, some of which were infected with *B. burgdorferi* but in contrast to *I. scapularis*, in which multiple life stages became infected, none of the *D. variabilis* ticks collected from the same population of mice were found infected with *B. burgdorferi* [35]. Thus, as a result of this rapid clearance of *B. burgdorferi*, *Dermacentor* fails to maintain a *B. burgdorferi* infection long enough for the bacteria to be transferred between developmental stages and this would be expected to prevent transmission to new hosts.

Nevertheless, some case studies provide indirect evidence that *Dermacentor* spp. ticks may be able to vector, as well as acquire, *Borrelia* infections. A case study from Bulgaria described an individual bitten by a *Dermacentor marginatus* tick, who subsequently developed an erythema migrans (EM) rash and was then found to be positive for *B. burgdorferi* exposure on serology, which was confirmed positive through a skin biopsy [74]. Similar anecdotal evidence of rashes and ulcers associated with *Dermacentor* spp. tick bites in France has also been noted [75]. Although these studies were on *Dermacentor* species other than *D. variabilis*, these results show that *Borrelia* can be detected in *Dermacentor* genus ticks and possibly vectored by them, which has reinvigorated investigation of the vectorial potential of *D. variabilis*. In a large study, 127 *D. variabilis* were obtained from the Human Tick Test Kit Program, created for ticks removed from military personnel in Wisconsin, USA. Of those ticks, 11% tested positive for *B. burgdorferi* by PCR [76]. These ticks were all engorged adults removed from human hosts, implying either that the *D. variabilis* ticks were infected with *B. burgdorferi* as nymphs and retained the infection into the adult stage or that the human hosts were infected prior to the tick bite and had a high enough spirochetal load to transmit the infection to the tick. This alternative was deemed unlikely as none had a known history of Lyme disease symptoms [76]. Although *D. variabilis* transiently acquiring *Borrelia* from an infected host does not equate to ready transmission of infection, there is a possibility, if not high probability of transmission if

the ticks refeed rapidly. Unambiguous evidence of the duration of viable *B. burgdorferi* persisting in *Dermacentor* spp. ticks would require culture of bacteria, collected at sequential intervals from the initial acquisition of infection. The persisting ambiguity about the ability of *D. variabilis* to acquire *Borrelia* spp. motivated our testing for *B. burgdorferi* and *B. miyamotoi* in *D. variabilis* ticks collected in New Brunswick. While no ticks met the formal criteria for *B. burgdorferi* or *B. miyamotoi* presence and both pathogens were under-represented in *D. variabilis* ticks relative to wild animals suggesting poor retention of *Borrelia* spp. these ticks are vectors for a variety of other pathogens that warrant surveillance and appear to be extending their range northward into New Brunswick.

## Study limitations and mitigations

This study took advantage of the power of community science; the public collected and submitted ticks throughout the region in exchange for identification of the tick and pathogen testing. Despite the many advantages of this community-based passive surveillance approach, interpretation of these findings entails awareness of issues intrinsic to passive surveillance. Passive surveillance relies on public detection and submissions of ticks, but the human population is not evenly distributed. As the tick recoveries documented here were not clustered around cities (Fig 2), using the location of tick encounters rather than donor address mitigated this bias. Motivation of members of public to participate and sample damage in the mail are difficult to quantify but are highly unlikely to be uniform, however, tick donations were primarily from veterinary clinics and participating veterinary clinics were largely constant during most of the ten-year span of this study. The first foci detected was in the southeast of the province, close to the study location so it is difficult to differentiate between the proximity to Nova Scotia's abundant *D. variabilis* populations from recovery bias in explaining this tick "hot spot". Nevertheless, the southwestern foci, which was not close to the study site, appeared only a year later, nor would recovery bias apply to the eTick platform. A bias affecting host data did occur in 2020. Prior to 2020, ticks from all hosts were collected and tested, whereas financial considerations resulted in paid testing in 2020 so that after this date, submissions were human-biased. As the eTick platform does not provide tick testing and involves no cost to the image contributors, this consideration would not apply to that data set.

## Conclusion

*Dermacentor variabilis*, the wood tick or American dog tick, is expanding its range across North America. This tick species is hardy, able to utilize diverse hosts and transmit pathogens of medical and agricultural importance. The 10-year span covered by this study documents the likely establishment of new population foci as the species expands northward. The pattern of population establishment appears responsive to both climactic factors and more random, but successful, "seeding" of populations in areas climactically sub-optimal for tick populations. As dogs were the most common host from which these ticks were recovered, human activity, via movement of companion animals, is strongly implicated as a direct driver of tick range expansion. This study spans the period during which this tick species appears to have colonized a new range and provides the opportunity to examine the dynamics of population and range expansion of an invasive tick species of medical and economic concern.

## Supporting information

**S1 File. Contains supporting table and figures.**
(PDF)

**S1 Raw images.**
(PDF)

## Acknowledgments

The authors would like to thank veterinary clinics and members of the public throughout the Maritimes who contributed specimens. We thank eTick for supplying data and Anne Berthold for comments on the manuscript.

## Author Contributions

**Conceptualization:** Vett K. Lloyd.

**Data curation:** Vett K. Lloyd.

**Formal analysis:** Vett K. Lloyd.

**Funding acquisition:** Vett K. Lloyd.

**Investigation:** Andrea M. Kirby, Ellis P. Evans, Samantha J. Bishop.

**Methodology:** Vett K. Lloyd.

**Project administration:** Vett K. Lloyd.

**Visualization:** Samantha J. Bishop.

**Writing – original draft:** Andrea M. Kirby, Ellis P. Evans, Vett K. Lloyd.

**Writing – review & editing:** Ellis P. Evans, Vett K. Lloyd.

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
