## [Decision Letter · Decision Letter 0]

26 Jul 2023

PONE-D-23-15954Establishment and range expansion of Dermacentor variabilis in the northern Maritimes of Canada: community participatory science documents establishment of an invasive tick speciesPLOS ONE

Dear Dr. Lloyd,

Thank you for submitting your manuscript to PLOS ONE. After careful consideration, we feel that it has merit but does not fully meet PLOS ONE’s publication criteria as it currently stands. Therefore, we invite you to submit a revised version of the manuscript that addresses all of the points raised during the review process. I concur with the reviewers that the section on screening for Borrelia burgdorferi and B. myamotoii is not necessary, and is actually distracting from the rest of the work. The manuscript describes that initial screening potentially identified B. burgdorferi, but sequencing of those amplicons showed them to not be B. burgdorferi. The one likely positive was unidentifiable, and its description suggests that it was not actually a Dermacentor.  The data indicate that there was no evidence of B. miyamotoii in any of the studied ticks. As written, readers will get the impression that those spirochetes were identified in your ticks: for both bacteria, the manuscript essentially says, "we thought that we had evidence of their presence, but it turned out that we were wrong". I strongly recommend that either the section on Borrelia be omitted, or the results be described in a straightforward manner. For example, "Ticks were screened by PCR and amplicon sequencing, but no evidence was found that any Dermacentor were infected with either spirochete."

We look forward to receiving your revised manuscript.

Kind regards,

Brian Stevenson, Ph.D.

Academic Editor

PLOS ONE

“The authors would like to thank veterinary clinics and members of the public throughout the Maritimes who contributed specimens. We thank eTick for supplying data and Anne Berthold for comments on the manuscript.

Seed funding for the tick bank was provided by the Canadian Lyme Disease Foundation, and operational funding was provided by the Natural Sciences and Engineering Research Council as well as private donations for tick research.”

“Seed funding for the tick bank was provided by the Canadian Lyme Disease Foundation to VKL (CanLyme 2014-1, https://canlyme.com/). Operational funding was provided by the Natural Sciences and Engineering Research Council to VKL (NSERC 4426-2015, https://www.nserc-crsng.gc.ca) as well as private donations for tick research. EPE was supported by a Undergraduate Student Research Award from the Natural Sciences and Engineering Research Council (NSERC USRA). The funders had no role in study design, data collection and analysis, decision to publish, or preparation of the manuscript.”

4. We note that Figure 2 and Supplemental Figure 3 in your submission contain [map/satellite] images which may be copyrighted. All PLOS content is published under the Creative Commons Attribution License (CC BY 4.0), which means that the manuscript, images, and Supporting Information files will be freely available online, and any third party is permitted to access, download, copy, distribute, and use these materials in any way, even commercially, with proper attribution. For these reasons, we cannot publish previously copyrighted maps or satellite images created using proprietary data, such as Google software (Google Maps, Street View, and Earth). For more information, see our copyright guidelines: http://journals.plos.org/plosone/s/licenses-and-copyright.

a. You may seek permission from the original copyright holder of Figure 2 and Supplemental Figure 3 to publish the content specifically under the CC BY 4.0 license. 

6. We notice that your supplementary figures and tables are included in the manuscript file. Please remove them and upload them with the file type 'Supporting Information'. Please ensure that each Supporting Information file has a legend listed in the manuscript after the references list.

Reviewers' comments:

Reviewer's Responses to Questions

**Comments to the Author**

1. Is the manuscript technically sound, and do the data support the conclusions?

Reviewer #1: No

Reviewer #2: Yes

Reviewer #3: Yes

2. Has the statistical analysis been performed appropriately and rigorously? 

Reviewer #1: No

Reviewer #2: Yes

Reviewer #3: N/A

3. Have the authors made all data underlying the findings in their manuscript fully available?

Reviewer #1: Yes

Reviewer #2: Yes

Reviewer #3: Yes

4. Is the manuscript presented in an intelligible fashion and written in standard English?

Reviewer #1: Yes

Reviewer #2: Yes

Reviewer #3: Yes

5. Review Comments to the Author

Reviewer #1: This may be a worthwhile study but requires some work before being acceptable for publication in my view.

The main issue is the simplistic treatment of the changes in numbers and geographic sources of ticks submitted in passive surveillance. Spatio-temporal changes in the proportions/populations of the public participating in passive surveillance need to be ruled out as an alternative explanation. This may be possible if the IDs of individual submitters is known, and an alternative is to ‘calibrate’ passive surveillance data against field surveillance data.

Introduction

In the introduction it is mentioned that D. variabilis is a vector of Anaplasma phagocytophilum (main vector I. scapularis and I. pacificus) and Ehrlichia chaffeensis (main vector A. americanum). There may be articles on vector competence of D. variabilis for these bacteria that I’m not aware of – if so they should be cited. However, I think, as for B. burgdorferi detections in these ticks, there are simply some articles that have found DNA in small numbers of the ticks, which may represent a past blood meal on a host infected with the bacteria rather than being evidence of vector competence (e.g. https://academic.oup.com/jme/article/40/4/534/999470). Therefore, it should be mentioned that while these bacteria have been found in D. variabilis, vector competency studies are needed to assess their role in transmission cycles, and their risk to human health. Similarly references should be provided for statements about vector competency for other pathogens cited.

When using scientific names to start a sentence, spell out the genus. Please use “these data” rather than “this data”.

The following section needs rewriting. “Suitable, if not optimal, climates for this tick species are present throughout the Canadian Maritimes and are predicted to increase in suitability with further climate change (Boorgula et al., 2020). Thus, it can be anticipated that D. variabilis will establish populations in New Brunswick which will subsequently expand. If the spread is limited by climate, the pattern of establishment will mimic that of Ixodes scapularis, a tick species that invaded the province earlier (Lieske & Lloyd, 2018; McPherson et al., 2017). In contrast, if the province is already suitable in climate, D. variabilis expansion into the region would be instead limited by introductions of the ticks.”

Boorgula et al. (2020) suggested that only southern New Brunswick was climatically suitable for D. variabilis. It is unlikely that climate change-driven range expansion of D. variabilis will precisely mimic that of I. scapularis as these ticks likely have rather different climatic thresholds for survival of their populations.

Methods

There is no mention of the keys or at least criteria used to identify the ticks to species and this needs to be mentioned.

As the PCR methodology has been previously published, and presuming that the method used was as published, I don’t think details of the primers (Table 1) and PCR cycle conditions are needed.

Results

That additional ticks were obtained from Geneticks and eTick needs to be mentioned in the methods rather than in results.

The demonstration of range expansion depends on by-eye examination of maps in figure 2. These maps are very small and difficult to view. The range expansion (by eye) is approximately 350km in 8 years, which would seem rapid. Apart from year-on-year increases in the numbers of ticks submitted (shown in figure 1), there is no form of statistical analysis or formal quantification of changes in geographic patterns and numbers of ticks. There is a need to rule out other possible causes of the apparent range expansion, which is that there is an increased participation by the public, over a wider are of New Brunswick, during the period of the study. At present it is an assumption that public participation was constant throughout the study.

The results of PCR testing are confusing. It is not clear why some ticks would test positive for Fla B and not Osp A, while for others the inverse was seen. Only 5 of 27 ticks tested for 16-23 IGS were positive. If B. burgdorferi were present in a tick surely all would be positive. I don’t see why results from H. leporispelustris ticks would be included here.

Discussion

The argument that abundant recovery of male ticks by 2017 supports existence of established populations, on the basis that “male ticks do not feed for long and so are less likely to be adventitious” is likely incorrect. Male and female adult ticks, if adventitious, would have most likely been dispersed as nymphal ticks.

The section on possible transmission of B. burgdorferi by D. variabilis is speculative and not helpful to the articles.

Reviewer #2: The manuscript by Kirby et al., present data on Establishment and range expansion of Dermacentor variabilis in the northern Maritimes of Canada. Overall, the manuscript is well described, and the information is interesting and important for researchers in the field of tick-borne diseases and ticks.

Only minor revisions are suggested to improve the manuscript.

The authors mentioned they conducted PCR to screen Dermacentor ticks for both B. burgdorferi and B. miyamotoi without doing a PCR to confirm and check tick DNA integrity. Please clarify? Also clarify if the tick is positive for Borrelia spp why some genes are positive, and some genes didn’t work with PCR?

Reviewer #3: I enjoyed this descriptive paper that focused on the ability for passive sampling and community science to document detection and range expansion of an invasive species. It is clear, and well-written. It also sheds light on a perceived ‘second rate’ tick species that receives much less attention than the blacklegged tick, but due to its potential medical importance, should be routinely monitored.

I mainly have minor comments that address typographical errors in the document.

Keywords: variabilis is misspelled

Last paragraph before the Materials and Methods: There is a sentence beginning with “This data” which likely should be “These data”

Under “Molecular Analysis” – there is an extra parenthesis before Libernardo

Table 2 – bold table title, and write genus name and province in full. I’d also recommend centering the column headers.

Some readers may be interested in how many D. albipictus you had in your submissions – do they account for a significant amount?

No description in methods of how you generated the maps in Fig 2 – I’d suggest including this.

The format of your subheadings changes between methods and results – pick one?

Figure 1 – this is minor, but there’s enough room on the x axis to write out the full names of the months; would that be possible to do? Also, it’s almost impossible to tell the difference between some of the more closely-shaded greys, making it hard to tell if the bars in March and April are from 2015, 2016, or 2017. Is it possible to add a hashing pattern or something to help distinguish? I’d also suggest that the figure caption should include some additional information on how the ticks were collected (e.g. passive submission).

In the results, one of my main points of consideration was that the increases over time could be related to increasing awareness of your tick services (which could also be correlated with the geographic spread of information away from the location of your tick centre). However, I think that you addressed this as a possible limitation, while also demonstrating that these are likely established populations. Thus, regardless of whether you do have a degree of sampling bias, it seems likely that your results overall demonstrate a new, and likely increasing, population of D. variabilis in NB.

Figure 2- the maps are a bit small; if the lines on the maps are differentiating the health zones, it is impossible to tell – is there any chance of increasing the size of the maps slightly and perhaps arranging them with 3 across? That would orphan one, but might help the size issue. Or, alternatively, if the health zones are important could you add numbers that can be seen more clearly? Finally, could you cite where these health zone designations come from?

On the next page after Fig 2 – I don’t think the genus for leporispalustris was given previously – write in full, if not.

Missing parenthesis after Peromyscus leucopus near bottom of a page in the discussion (no page #s!)

Suggest including a reference for “male ticks do not feed for long…”

Grubhoffe and Hynes references are italicized

Something a bit grammatically strange in the sentence “An adult D. albipictus was found…” – read through again? Perhaps remove the “was”?

Suggest starting a new paragraphs at “While there is evidence of D. variabilis carrying…”

6. PLOS authors have the option to publish the peer review history of their article (what does this mean?). If published, this will include your full peer review and any attached files.

Reviewer #1: No

Reviewer #2: No

Reviewer #3: No

---

## [Author Response · Author response to Decision Letter 0]

9 Sep 2023

Dear Dr. Stevenston,

Thank you for your managing our manuscript “Establishment and range expansion of Dermacentor variabilis in the northern Maritimes of Canada: community participatory science documents establishment of an invasive tick species”. We are delighted that the data on the range expansion of D. variabilis was felt to be of value to the research community. We would like to thank you and all the Reviewers for your and their time in reading the manuscript and providing such insightful comments. It is a rare to get such penetrating and constructive comments and we feel the manuscript is very much improved as a result. 

We have made all the corrections and changes requested, with the exception of not completely deleting but merely greatly shortening the section on Borrelia spp. in D. variabilis (rationale for this decision is detailed below). The clean revised and track-changed version of the manuscripts have been uploaded with this submission, along with the supplemental material and original gel images.

A detailed explanation and tracking of responses are provided below. The most significant changes are: 

• For the issue of whether there Borrelia spp. are found in D. variabilis ticks, I would prefer to retain that information - it is an important question from a public health perspective and common question from the public as adult D. variabilis readily feed on humans. This was the original motivation for addressing this question. This work also represented a year's worth of work for each of the first two student authors. And although negative data doesn’t make for exciting reading, it is still of value and B. miyamotoi remains understudied. However, I completely agree that the previous narrative style was an unnecessarily diffuse and confusing way to present the information. To address this problem, I have greatly shortened this section. I have also firmly placed it within the context of the difference between a tick acquiring a pathogen and a tick being able to vector the pathogen. As this section is now more concise, and, I believe, clearer and addresses a point of public health importance, albeit with negative data, I would argue that its inclusion remains of value. 

• The distinction between a tick being able to acquire a pathogen, Borrelia spp. or others, and being able to transmit it is now made much more explicitly and reiterated in the introduction, results and discussion. 

• The question of whether increased recoveries of D. variabilis represent increased presence or increased search effort is now addressed much more explicitly in the MM, Results and Discussion. Statistical analysis, albeit simple statistical analysis due to the nature of the data, is now presented in the Results to address the question of range expansion versus increased surveillance effort. 

• We have reformatted the maps to make them easier to view, as suggested by both reviewers. While the current format was the one suggested by Reviewer 3, the format that allows for the largest individual panels is a landscape orientation. This variation is also in the submitted material, should it be useful. 

• We have rewritten the portion of the introduction and discussion that addresses the driver(s) of tick range expansion to emphasize the hypothesized anthropic aspect proposed here. 

• We have clarified the PCR methodology, both details and limitations, in the MM, Results and Discussion. 

We have addressed the comments of the Reviewers and details of these changes are provided below, and in the track-changed manuscript. Again, we express our appreciation for the time and effort taken both by the Editor and the Reviewers in reviewing the manuscript. We always appreciate the time and suggestions of Reviewers but these reviews were exceptionally penetrating and thoughtful and we feel that addressing them has made for a stronger work. 

The reviewer comments are in italics and our responses are inter-digitated below each. Line numbers refer to the track-changed version of the manuscript. 

Again with our thanks,

Vett Lloyd (for all authors) 

Detailed responses to Editorial notes and Reviewers 

Editorial Issues: 

1. I believe that the manuscript is now formatted correctly, or at least more correctly. 

2. Acknowledgments: We have removed the funding acknowledgements from the “Acknowledgements”. I had left them there because I was initially unable to enter the Canadian Lyme disease Foundation (CanLyme) in the funding section drop down menu, however, the funding statement is correct and we have removed the duplicate information from the acknowledgements.

3. “Data not shown” item: This phrase was removed – in retrospect, its inclusion was not helpful – there were no sequences, so nothing was available to show. The statement “the presence of B. miyamotoi could not be confirmed by Sanger sequencing” conveys that information much more appropriately. 

4. Use of GIS template maps. We have looked into the copyright issue for the map templates we used and have confirmed that academic use of these templates is permitted. The information page ( - https://doc.arcgis.com/en/arcgis-online/reference/display-copyrights.htm) states the following: 

Terms of use for static maps

The following uses are permitted:

• Personal use, internal business use, or to include in a presentation or a report for a client

• In brochures and marketing collateral, or on a company website to promote your own products and services and display your store locations

• In academic publications (for example, research journals, textbooks, and so on)

• In government works, so long as the government agency clearly delineates between government works that are in the public domain and third party works that are protected by copyright. The following attribution is recommended as a caption to the image: Map image is the intellectual property of Esri and is used herein under license. Copyright © 2020 Esri and its licensors. All rights reserved.

We have amended the image captions and the MM to include this information – and I apologize for not researching this prior to submission. I had assumed that use of the templates were permitted, but hadn’t researched it sufficiently to be certain at the time of submission, which is why I indicated that there might be copyright issues. I have now confirmed that there are no copyright issues, to the best of my ability to determine. 

5. Original images of the gels in the supplemental material are provided.

6. The supporting information has been removed from the main manuscript and is now a separate file. 

Reviewer: 1

General comment: 

The main issue is the simplistic treatment of the changes in numbers and geographic sources of ticks submitted in passive surveillance. 

Apart from year-on-year increases in the numbers of ticks submitted (shown in figure 1), there is no form of statistical analysis or formal quantification of changes in geographic patterns and numbers of ticks. There is a need to rule out other possible causes of the apparent range expansion, which is that there is an increased participation by the public, over a wider are of New Brunswick, during the period of the study. At present it is an assumption that public participation was constant throughout the study.

The demonstration of range expansion depends on by-eye examination of maps in figure 2. These maps are very small and difficult to view. 

This is an important issue which we had addressed only in passing in the discussion (limitations and mitigations). We have now addressed this question more explicitly in the Results section (with concomitant details in the MM). 

Spatio-temporal changes in the proportions/populations of the public participating in passive surveillance need to be ruled out as an alternative explanation. This may be possible if the IDs of individual submitters is known, and an alternative is to ‘calibrate’ passive surveillance data against field surveillance data.: 

Search effort: It is difficult to rule out increased recoveries based on knowing the IDs of participants; the ticks were submitted by NB veterinary clinics and while we do have records of the names of tick donors, this is not particularly meaningful. That said, outreach to veterinary clinics occurred in 2012 and 2013 (by letter/email and presentation at the provincial veterinary medicine conference) when the tick bank initiative was started. There was no organized recruitment of clinics after that point (ie we responded to calls from vet clinics asking about tick testing but we did not approach any clinics). Clinic engagement remained constant through the duration of the study with the exception of loss of clinics starting in ∼ 2017/2018 when private clinics were bought out by a veterinary medicine consortium; the consortium did not appear to support free services so submissions from some clinics were lost. This change would have resulted in under-reporting of tick recoveries, however, rather than over-reporting. 

Action: A briefer version of the recruitment strategy has now been added to the MM and this elaboration on the point of search effort has been added to the section “study limitations and mitigations” in the discussion (Lines 169-175 and 592-594).

Calibration against active surveillance: Active surveillance for D. variabilis would be the ideal calibration. Unfortunately, I’m not award of past active surveillance that detected D. variabilis in this region and initiating active surveillance would be beyond the scope of this study, although the value of doing so is noted (Line 414). Gabriele-Rivet et al (https://journals.plos.org/plosone/article?id=10.1371/journal.pone.0131282) did conduct active surveillance in the province in 2014 and did not detect any D. variabilis. This isn’t overly surprising as during this year only ∼ 1% of ticks recovered by passive surveillance were D. variabilis. 

Action: This is a valuable point to explicitly note, and we have done so (Lines 414-417). 

Statistical analysis of range expansion: Providing a quantitative correlate to the visual presentation of the range expansion data we did a simple statistical analysis of the data to demonstrate that that numbers of D. variabilis recovered increased over time and also was over-represented from the north and middle of the province (due to the dearth of I. scapularis recoveries from these regions – the absolute numbers of D. variabilis recovered from the southern regions was greater than in the middle of the north of the province. 

Action: Explanation on the application of the Chi squared testing is provided in the MM (Lines 202-213) and results are now mentioned in the Results section (Lines 293-295 and 313-318). The size and presentation of the maps have been improved. 

In the introduction it is mentioned that D. variabilis is a vector of Anaplasma phagocytophilum (main vector I. scapularis and I. pacificus) and Ehrlichia chaffeensis (main vector A. americanum). There may be articles on vector competence of D. variabilis for these bacteria that I’m not aware of – if so they should be cited. However, I think, as for B. burgdorferi detections in these ticks, there are simply some articles that have found DNA in small numbers of the ticks, which may represent a past blood meal on a host infected with the bacteria rather than being evidence of vector competence (e.g. https://academic.oup.com/jme/article/40/4/534/999470). Therefore, it should be mentioned that while these bacteria have been found in D. variabilis, vector competency studies are needed to assess their role in transmission cycles, and their risk to human health. Similarly references should be provided for statements about vector competency for other pathogens cited.

Thank you very much for these comments. We had elided a description of known and suspected pathogens. With reorganization and clarification of this paragraph, the presentation of this information is much clearer. 

Action: This paragraph has been reorganized to a) provide full primary citations b) separate the pathogens for which vector competence has been confirmed from those where the pathogens may simply have been acquired from infected hosts and c) explicitly point out this distinction and the need for experimental confirmation of vector competence (Lines 87-121). 

The following section needs rewriting. “Suitable, if not optimal, climates for this tick species are present throughout the Canadian Maritimes and are predicted to increase in suitability with further climate change (Boorgula et al., 2020). Thus, it can be anticipated that D. variabilis will establish populations in New Brunswick which will subsequently expand. If the spread is limited by climate, the pattern of establishment will mimic that of Ixodes scapularis, a tick species that invaded the province earlier (Lieske & Lloyd, 2018; McPherson et al., 2017). In contrast, if the province is already suitable in climate, D. variabilis expansion into the region would be instead limited by introductions of the ticks.”

Boorgula et al. (2020) suggested that only southern New Brunswick was climatically suitable for D. variabilis. It is unlikely that climate change-driven range expansion of D. variabilis will precisely mimic that of I. scapularis as these ticks likely have rather different climatic thresholds for survival of their populations.

Thank you again for this excellent comment. This paragraph very much needed rewriting for clarity. The original intention of this paragraph was to set up a (simplistic) set of hypotheses for the source of D. variabilis range expansion – natural vs anthropic forces, although the biology is always more complicated, as noted. This intent was compromised by less than clear writing. 

Action: This paragraph has been rewritten and broken into two paragraphs to clarify the attempt to distinguish between anthropic and natural drivers of introduction and range expansion (Lines137-163). The reference to the climactic suitability has been amended (Line 138)– thank you. The comparison between I. scapularis and D. variabilis range expansion has been clarified – the point was the role of climatic abiotic forces rather than expecting the two tick species to follow the same pattern of expansion. This, however, was not what was stated in the original version. Thank you again for flagging this. 

As the PCR methodology has been previously published, and presuming that the method used was as published, I don’t think details of the primers (Table 1) and PCR cycle conditions are needed.

The details of PCR cycles have been deleted (Lines 239-249). Some of the primers used, but not all, are described in the given citation so we retained Table 1, which lists all primers used. 

The results of PCR testing are confusing. It is not clear why some ticks would test positive for Fla B and not Osp A, while for others the inverse was seen. Only 5 of 27 ticks tested for 16-23 IGS were positive. If B. burgdorferi were present in a tick surely all would be positive. 

This point was also flagged by Reviewers 2 and 3. This is most likely a problem causes by a low number target copies relative to tick or host DNA as we see it fairly commonly in PCR of non-engorged ticks (regardless of species). Additional explanation is provided in the MM (Lines 224-228 and Discussion Lines 493-499). 

The section on possible transmission of B. burgdorferi by D. variabilis is speculative and not helpful to the articles.

The value of the Borrelia testing section was also flagged by Reviewer 3 and the Editor. The observation that it is tangential to the main point of the study – D. variabilis range expansion – is entirely correct. Nevertheless, this is a question that comes up frequently as the tick species is often found feeding from human hosts, which is why we addressed it. As it represented substantial work by two of the student authors, I would prefer to retain this section, albeit in a very much more succinct form and with the distinction between sporadic acquisition of pathogen from a host vs vector capacity made very clear. 

Action: This section has been greatly truncated (and the information on other tick species deleted – Lines 335 - 353). The distinction between acquisition of Borrelia spp. vs transmission is much more clearly delineated in the introduction and discussion (Lines 106-107, 509-510).

.

The argument that abundant recovery of male ticks by 2017 supports existence of established populations, on the basis that “male ticks do not feed for long and so are less likely to be adventitious” is likely incorrect. Male and female adult ticks, if adventitious, would have most likely been dispersed as nymphal ticks.

This is a good point, also flagged by Reviewer 3. While larval and nymphal D. variabilis are typically found feeding from small mammals and rarely from birds or larger mammals (ie doi: 10.1093/jmedent/33.3.379, doi: 10.1093/jmedent/32.4.453, doi: 10.1093/jmedent/30.4.740, doi: 10.1139/z79-258, doi: 10.1139/z78-004), which would limit their dispersal as immatures, the overall point of extrapolating from tick sex to the source of their acquisition was weak and peripheral, and has now been deleted (Lines 399-405). 

The range expansion (by eye) is approximately 350km in 8 years, which would seem rapid. 

This is a very good point and supports the proposal that the dispersal of this tick species is driven by anthropic forces – presumably quite literally. This is now noted in the discussion (Lines 446-448) 

I don’t see why results from H. leporispelustris ticks would be included here.

Agreed – this is a distraction, inconclusive and is now deleted (Line 344). 

Reviewer #2: The manuscript by Kirby et al., present data on Establishment and range expansion of Dermacentor variabilis in the northern Maritimes of Canada. Overall, the manuscript is well described, and the information is interesting and important for researchers in the field of tick-borne diseases and ticks.

Thank you

The authors mentioned they conducted PCR to screen Dermacentor ticks for both B. burgdorferi and B. miyamotoi without doing a PCR to confirm and check tick DNA integrity. Please clarify? 

This was poorly phrased on our part. At the time of initial testing of the ticks, the extracted DNA was checked for integrity to confirm that the DNA extraction generated product suitable for analysis. These DNA samples were then stored at -20oC for future analysis, including re-testing of putative Borrelia-positive Dermacentor ticks. During this time some DNA samples appear to have degraded, possibly due to freezer malfunction or thawing during routine freezer cleaning. This point is now mentioned in the text (Line 496-499). 

Also clarify if the tick is positive for Borrelia spp why some genes are positive, and some genes didn’t work with PCR?

This is a good point flagged by the other reviewers as well. This arises, presumably, from a sensitivity issue in picking out the target Borrelia sequence in the context of more abundant vector DNA. This point has been added to the MM and is expanded upon in the Discussion (Lines 224-228 and 493-495). 

Reviewer #3: I enjoyed this descriptive paper that focused on the ability for passive sampling and community science to document detection and range expansion of an invasive species. It is clear, and well-written. It also sheds light on a perceived ‘second rate’ tick species that receives much less attention than the blacklegged tick, but due to its potential medical importance, should be routinely monitored.

Thank you! 

Some readers may be interested in how many D. albipictus you had in your submissions – do they account for a significant amount?

Few D. albipictus were recovered as the tick bank primarily receives ticks from pets and only occasionally will a hunter submit ticks from a moose or their hunting dogs. We received 13 over the 10 years of the study, but these represented 4 from 1 moose carcass in 2015 (there were presumably more that were not collected from the carcass), 7 from a single hunting dog associated with that moose hunt, and 2 others, each from different dogs in 2019. A sentence of information on D. albipictus has been added to the results section (Line 268).

In the results, one of my main points of consideration was that the increases over time could be related to increasing awareness of your tick services (which could also be correlated with the geographic spread of information away from the location of your tick centre). However, I think that you addressed this as a possible limitation, while also demonstrating that these are likely established populations. Thus, regardless of whether you do have a degree of sampling bias, it seems likely that your results overall demonstrate a new, and likely increasing, population of D. variabilis in NB.

Thank you for this note – distinguishing between recoveries due to increased “search effort” versus increased tick numbers, was a concern also noted by Reviewer 1. We have tried to address this point by providing more information on the recruitment strategy in the MM and further elaboration on this point in the Discussion section “study limitations and mitigations” (Lines 169-175 and 592-594). We also provide a simple statistical testing of D. variabilis population increases in the MM (Lines 202-213) and results (Lines 293-295 and 313-318). 

Keywords: variabilis is misspelled

Thank you. I believe that this will need to be corrected on the manuscript submission platform, which I hope that we can do ourselves, otherwise I will reach out to the editorial office for assistance. Thank you for noting this!

Figure 2- the maps are a bit small; if the lines on the maps are differentiating the health zones, it is impossible to tell – is there any chance of increasing the size of the maps slightly and perhaps arranging them with 3 across? That would orphan one, but might help the size issue. Or, alternatively, if the health zones are important could you add numbers that can be seen more clearly? 

Finally, could you cite where these health zone designations come from?

Thank you for these suggestions. The 3X3+1 format works quite well – I was initially concerned that the orphaned year would look strange, but it works well. I have modified both maps, Figure 2 and Supplemental Figure 3 in this way. I also found that by making a horizontal rather than vertical figure, each panel could be further expanded. I will include these as alternate formats when submitting as changing from portrait to landscape in the manuscript could be disruptive. 

The increased size of the panels should make the health zones larger. They have little intrinsic meaning other than they are supposed to represent approximately equal human population groups. With the addition of analysis of population-based recovery to the manuscript, which alludes to these zones, depicting them now makes more sense. A citation for these zones is now provided (Lines 208-210). 

Suggest including a reference for “male ticks do not feed for long…”

This point has been removed – as noted by Reviewer 1, the argument is not strong I had elided repletion with feeding time so the argument that recovery of male ticks could be used as an indicator of established populations became too weak to include (Lines 399-401). We thank the reviewers for raising this point!

Corrections relating to the minor comments are listed below. 

Minor Comments: These have been addressed. We thank the reviewers for noting them, with some embarrassment for not having noted the typographical errors (and error of grammar in the use of “data”) ourselves. We are very grateful for the reviewers’ thoroughness in reading and reviewing the manuscript. 

R1 When using scientific names to start a sentence, spell out the genus. 

R1 and R3 Please use “these data” rather than “this data”.

R1 There is no mention of the keys or at least criteria used to identify the ticks to species and this needs to be mentioned (Line 177). 

R1 That additional ticks were obtained from Geneticks and eTick needs to be mentioned in the methods rather than in results (Lines 192-195).

R3 Under “Molecular Analysis” – there is an extra parenthesis before Libernardo

R3 Table 2 – bold table title, and write genus name and province in full. I’d also recommend centering the column headers.

R3 No description in methods of how you generated the maps in Fig 2 – I’d suggest including this.(Added – Lines 199-201)

R3 The format of your subheadings changes between methods and results – pick one?

F3 Figure 1 – this is minor, but there’s enough room on the x axis to write out the full names of the months; would that be possible to do? Also, it’s almost impossible to tell the difference between some of the more closely-shaded greys, making it hard to tell if the bars in March and April are from 2015, 2016, or 2017. Is it possible to add a hashing pattern or something to help distinguish? I’d also suggest that the figure caption should include some additional information on how the ticks were collected (e.g. passive submission).

R3 On the next page after Fig 2 – I don’t think the genus for leporispalustris was given previously – write in full, if not. (Sorry – this is now deleted information – Line 344)

R3 Missing parenthesis after Peromyscus leucopus near bottom of a page in the discussion 

R3 (no page #s!)- Sorry about that!

R3 Grubhoffe and Hynes references are italicized

R3 Something a bit grammatically strange in the sentence “An adult D. albipictus was found…” – read through again? Perhaps remove the “was”?

R3 Suggest starting a new paragraphs at “While there is evidence of D. variabilis carrying…”

---

## [Editor Report · Decision Letter 1]

27 Sep 2023

Establishment and range expansion of Dermacentor variabilis in the northern Maritimes of Canada: community participatory science documents establishment of an invasive tick species

PONE-D-23-15954R1

Dear Dr. Lloyd,

We’re pleased to inform you that your manuscript has been judged scientifically suitable for publication and will be formally accepted for publication once it meets all outstanding technical requirements.

Kind regards,

Brian Stevenson, Ph.D.

Academic Editor

PLOS ONE
---

## [Editor Report · Acceptance letter]

5 Oct 2023

PONE-D-23-15954R1 

Establishment and range expansion of *Dermacentor variabilis* in the northern Maritimes of Canada: community participatory science documents establishment of an invasive tick species 

Dear Dr. Lloyd:

I'm pleased to inform you that your manuscript has been deemed suitable for publication in PLOS ONE. Congratulations! Your manuscript is now with our production department. 

Kind regards, 

on behalf of

Prof. Brian Stevenson 

Academic Editor

PLOS ONE